# EHRFlow: A Large Language Model-Driven Iterative Multi-Agent Electronic Health Record Data Analysis Workflow

Hao Wu*
Peking University
Beijing, China
xhawborn@gmail.com

Yinghao Zhu*
Peking University
Beijing, China
yhzhu99@gmail.com

Zixiang Wang
Peking University
Beijing, China
zixiangwang@buaa.edu.cn

Xiaochen Zheng
ETH Zürich
Zürich, Switzerland
xzheng@ethz.ch

Ling Wang
Affiliated Xuzhou
Municipal Hospital of
Xuzhou Medical University
Xuzhou, Jiangsu, China

Wen Tang
Peking University Third
Hospital
Beijing, China

Yasha Wang
Peking University
Beijing, China
wangyasha@pku.edu.cn

Chengwei Pan
Beihang University
Beijing, China
pancw@buaa.edu.cn

Ewen M. Harrison
University of Edinburgh
Edinburgh, Scotland, UK
ewen.harrison@ed.ac.uk

Junyi Gao†
University of Edinburgh
Health Data Research UK
Edinburgh, Scotland, UK
junyi.gao@ed.ac.uk

Liantao Ma†
Peking University
Beijing, China
malt@pku.edu.cn

## ABSTRACT

EHRFlow is a large language model-driven platform that simplifies electronic health record (EHR) data analysis for physicians through natural language interactions, eliminating the need for complex coding. EHRFlow integrates four key agents: PlanAgent, ToolAgent, CodeAgent, and ReviewAgent in a dual-loop mechanism that efficiently decomposes and executes tasks via PlanAgent, from coarse-grained conceptual planning to fine-grained task execution, leveraging a healthcare tool bank and the code generation and execution capabilities of ToolAgent and CodeAgent. The ReviewAgent ensures the safety and quality of code through iterative feedback and refinement. EHRFlow offers a user-friendly solution for physicians' customized needs while preserving sensitive data's privacy by executing code operations locally. Demonstrated through case studies, EHRFlow showcases its ability to coordinate agents in resolving complex medical data analysis tasks, highlighting its flexibility, scalability, and efficiency. The code for EHRFlow is publicly released at https://github.com/PKU-AICare/EHRFlow.

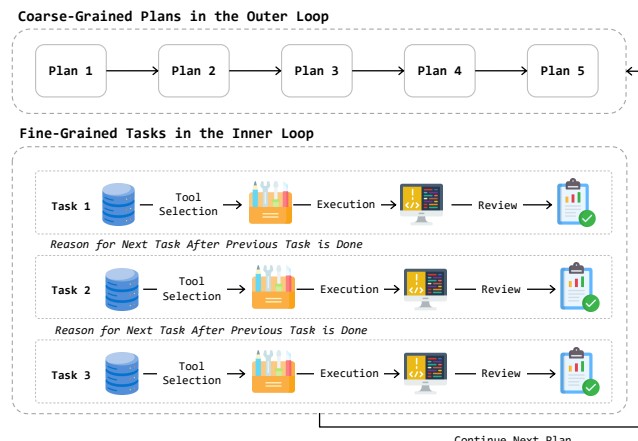

**Figure 1: Illustration of plan and task decomposition and execution pipeline in the loop.**

## 1 INTRODUCTION

Electronic health records (EHRs) are comprehensive digital repositories of health information [1, 6, 12]. Leveraging machine learning and deep learning approaches, valuable knowledge can be mined from EHR data to aid physicians in analyzing patients' health conditions. These approaches support healthcare in various ways, including personalized treatments, disease prediction, clinical research, public health surveillance, and resource allocation [5, 23, 24]. Despite the potential benefits, the development of AI models and the analysis of health data are typically conducted by health data scientists with a computer science background. Physicians, on the other hand, often have limited skills in complex EHR data analysis. Consequently, effective EHR data analysis requires close collaboration and frequent communication between physicians and AI experts. Physicians have expressed significant interest in analyzing their specific domain EHR data to enhance their clinical decision-making processes [11]. However, the skill gap between physicians and data scientists presents a challenge.

This gap motivates us to develop a platform that allows physicians to enter their data analysis requirements in natural language. We envision that this platform could interpret these requirements and execute the corresponding data analysis tasks automatically, without human intervention. As indicated in [4], physicians expect the digital agents to integrate into their clinical workflows through interactive conversations, expecting these AI tools to enhance their work efficiency and effectiveness.

---

*Equal contribution.
†Corresponding author.

The recent advent of large language models (LLMs) [14] with their instruction-following, reasoning, and function-calling capacities, has made implementing such a platform possible. Specifically, building upon the LLM's capacities, the LLM agent could form an autonomous proxy system that integrates other tools and technologies to enhance model performance and execute tasks efficiently[19]. For example, AutoGPT [7] showcases its capability in calling functions and executing code to achieve users' software development use cases, such as building a front-end website. However, such LLM agent systems cannot be directly applied in the medical domain where physicians have much more customized needs and privacy concerns. The challenges include: (1) For customized and professional use cases, LLM cannot understand the inherent processes without prompting the LLM with domain-specific knowledge. (2) Healthcare tasks are sometimes quite complicated and require precise and rigorous execution results, while conventional agent systems are prone to collapse. (3) Privacy matters in healthcare data, posing challenges in designing a privacy-preserved approach so that no patient data is uploaded when calling LLM's online APIs.

Aiming to address these challenges, we introduce EHRFlow, an iterative multi-agent system supported by LLMs, designed to serve for physicians' diverse EHR analysis requirements in their clinical workflows. The web interface is demonstrated in Figure 5. As illustrated in Figure 2, EHRFlow consists of four main agents: PlanAgent, ToolAgent, CodeAgent, and ReviewAgent. Inspired by XAgent [17], we refine the dual-loop mechanism of task decomposition and dynamic execution in the healthcare domain. The outer loop conceptualizes complex tasks into plans, and each plan is further split into lower-level tasks by the PlanAgent one by one. The inner loop refines and executes the task using tools retrieved from the healthcare tool bank, leveraging ToolAgent's reasoning and CodeAgent's code generation and execution capabilities. After finishing the task, PlanAgent continues to reason for the next task, as demonstrated in Figure 1. Meanwhile, the ReviewAgent examines the execution results and provides reflection and feedback to the PlanAgent and ToolAgent. Moreover, EHRFlow protects patients' privacy by only providing data schema, and all sensitive data-related operations are executed locally.

EHRFlow's contributions can be summarized as follows:

(1) We develop and deploy EHRFlow to assist physicians' data analysis requirements through simple natural language interactions, lowering the threshold with no need to write code. EHRFlow has a user-friendly interface, meets the needs of professional clinical use cases, and is scalable for more customized use cases by registering tool APIs in the tool bank and providing usage illustrations.

(2) EHRFlow tailors a multi-agent dual-loop mechanism of coarse-grained and fine-grained task decomposition specifically designed for the healthcare domain. This mechanism ensures effective management and optimization of complex medical tasks with privacy protection through the concept decomposition and sub-plan generation of both loops, as well as reviewing, reflection, and feedback correction within the each loop.

(3) Code for the tool banks and EHRFlow has been publicly released, offering transparent and reliable resources for the health data science community.

## 2 RELATED WORK

### 2.1 Healthcare Data Analytics Toolkits

Open-source projects like PyHealth [20] and AutoPrognosis [10] are two Python toolkits specifically designed for EHR data, simplifies deep learning integration for healthcare data analysis, supporting various data modalities and models. PyHealth focuses on clinical predictive modeling, mainly designed for both ML researchers and medical practitioners, while AutoPrognosis additionally offers more customized model explanations and cohort analysis features.

However, the gap remains between physicians and code-based data analysis workflow, a seamless data analysis and physician-driven data decisions require further development, a lower-threshold workflow is still to be implemented [4].

### 2.2 LLM-Driven Agent-Based Data Analytics

As Large Language Models (LLMs) have demonstrated significant intelligence, leveraging them as planning modules for autonomous agents has gained increased attention [9]. This paradigm involves generating both reasoning traces and task-specific actions in an interleaved manner, allowing for greater synergy between the two. Reasoning traces help the model induce, track, and update action plans as well as handle exceptions, while actions enable it to interface with external sources, such as knowledge bases or environments, to gather additional information [21].

Building on this foundation, multiple agents in a collaborative environment, each with distinct attributes and roles, can handle larger-scale and more complex tasks [16]. For example, TaskWeaver [13] introduces stateful execution and code verification to validate generated code before execution, thereby preventing potential issues. Similarly, XAgent incorporates reflection to provide feedback to other agents, which is useful for identifying bugs and making refinements [17].

In the healthcare domain, EHRAgent [15] automates EHR multi-table inference tasks from natural language instructions, simplifying interactions for healthcare workers with EHR systems. Additionally, Cui et al. [2] explore the feasibility of LLMs in few-shot EHR disease prediction, proposing frameworks with predictive and critical agents for narrative generation and error feedback to improve accuracy. However, existing LLM agent work in EHR data has limitations, particularly in tool usage, as it primarily relies on the LLM's own reasoning capabilities.

## 3 METHODOLOGY

Figure 2 illustrates the overall framework of EHRFlow, showcasing its hierarchical dual-loop processes. The outer loop is responsible for coarse-grained planning, while the inner loop focuses on fine-grained task execution.

### 3.1 Conceptual Decomposition-Centric Reasoning, Planning and Code Execution

User requirements typically involve multiple interrelated entities and concepts, forming a complex semantic concept network. Although large language models have high capabilities in understanding intent and reasoning, they still face challenges in dealing with these complex coupled conceptual systems. At the understanding

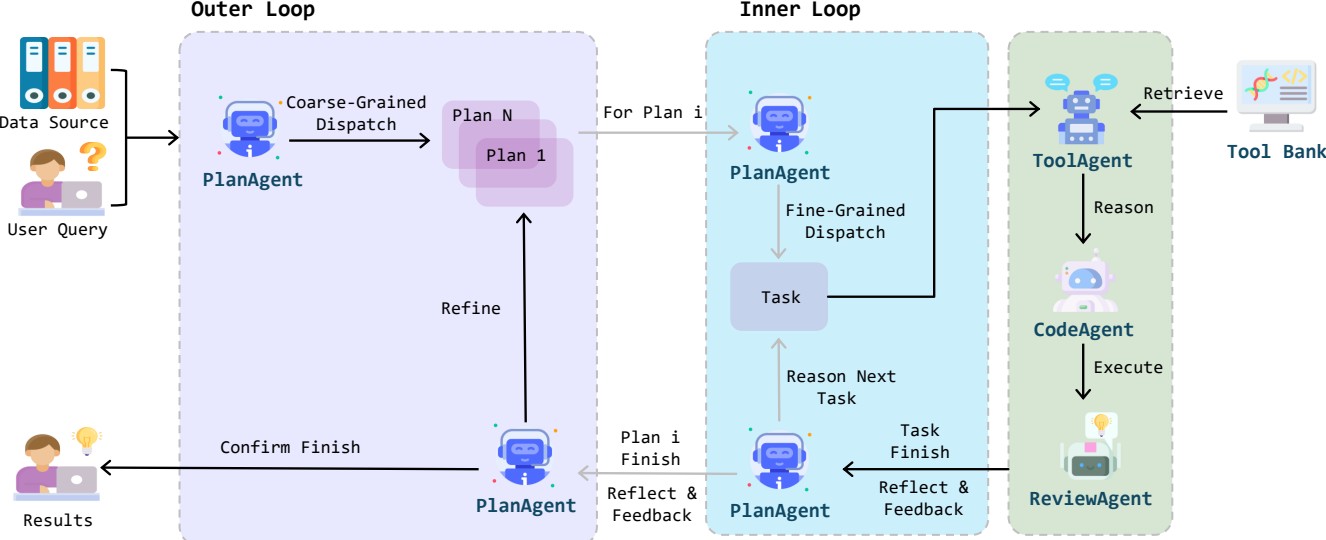

**Figure 2: Overall framework of EHRFlow. EHRFlow accepts user's natural language queries, and consists of the hierarchical dual-loop processes: the outer loop for coarse-grained planning and the inner loop mainly for fine-grained task executing.**

level, the complexity of coupled concepts may make it difficult for the model to quickly identify relationships and meanings between entities, especially when the requirements are unclear or the concept boundaries are fuzzy. At the execution level, transforming abstract concepts into concrete solutions is a challenge, especially in scenarios requiring the integration of cross-domain knowledge and multi-step logical reasoning.

Inspired by the Chain of Thought (CoT) [18] and Rephrase and Refine (RaR) [3] prompt strategies, targeted prompts are designed to guide the model in presenting its reasoning overview. This process includes steps such as question restatement, key concept identification, conceptual decomposition, reflection, deep thinking, reasoning, and action planning. EHRFlow adapts the decomposition of medical tasks into dual granularity levels from coarse in the outer loop to fine in the inner loop. As shown in Figure 1, the PlanAgent decomposes the user query instruction into multiple coarse-grained plans in the outer loop. Within the inner loop of each plan, the PlanAgent further decomposes into the fine-grained executable task, where ToolAgent reasons for choosing required tool APIs from the tool bank and the CodeAgent generates glue code and executes the code given the function calling of healthcare tool interfaces. After completing code execution, ReviewAgent checks the current status, reflects, and provides feedback to PlanAgent. If everything is on track, then PlanAgent continues to reason for the next task towards the target of the current plan.

During the above process, the prompt for each component also follows the strategy of concept identification and decomposition, reflection, reasoning, etc., and is adjusted appropriately based on the complexity of the tasks to support customized diverse medical task decomposition and arrangement. Our elaborately designed prompts ensure that the system can flexibly handle tasks of various complexities, optimizing the entire decision-making process from macro planning to micro execution.

Note that during the execution of tasks, the CodeAgent continuously maintains the code execution status throughout the dialogue process, similar to the programming mode in Jupyter Notebook [13], ensuring the coherence and consistency of code execution.

Furthermore, due to the sensitive nature of healthcare data, during the agent reasoning process, the context only includes the data descriptions, i.e., the column names indicating all features, and the data recorded format. The large language model has the ability to generate the correct code with these data descriptions as background for data analysis operations.

## 3.2 Reflection and Feedback-Based Dynamic Execution Mechanism

To address the challenges posed by the complexity of the environment and the potential issues of safety, harmlessness, and executability in the code generated by large language models for EHR data analysis, EHRFlow introduces a ReviewAgent to establish a reflection and feedback phase. This mechanism is designed to monitor each step's execution results and dynamically adjust based on the differences between actual and expected outcomes.

Speficically, the PlanAgent in the outer loop acts as a project manager, reviewing the execution results of sub-tasks and adjusting planned steps based on feedback. This aims to address issues and anomalies during execution, avoid fault loops, and seek alternative solutions, thereby facilitating the smooth progress of tasks. Additionally, the ReviewAgent conducts a comprehensive review after code generation, screening for harmful information, verifying the code's executability, evaluating its alignment with user requirements, reviewing code structure and style, and checking for exception handling and error handling mechanisms.

In the event that the code fails the review, the ReviewAgent provides clear review results and feedback to the large language

model, initiating a new cycle of code generation and review. This iterative process continues until the code fully meets the required standards, aiming to improve code quality, enhance trustworthiness, and provide doctors with a more reliable data analysis tool.

By obtaining perceptual information through real-time interaction and combining it with actual trial results, EHRFlow's feedback loop is designed to enable the workflow to quickly self-correct when facing inefficient planning or continuous errors. This transition from relying on preset perceptual information to real-time dynamic adjustment seeks to enhance the comprehensiveness and effectiveness of planning, strengthen adaptability and robustness in dynamic environments, and ultimately enable large language models to make more effective decisions and responses for new complex tasks.

### 3.3 Retrieval-Enhanced Healthcare Analysis Tool Bank Design

In the process of using EHR datasets to assist medical decision-making, we face challenges due to the inherent complexity of the data and the need for in-depth analysis. Therefore, meticulous and customized preprocessing of EHR data is essential to ensure that the data quality meets the standards for advanced analysis. Although EHR data is commonly stored in CSV file format, basic processing of CSV data alone is insufficient to meet the requirements of complex analysis. This approach can lead to misunderstandings and misjudgments of the information. Hence, a series of steps including data cleaning, transformation, integration, and advanced analysis are necessary to accurately extract and present the key information in EHR data. Additionally, existing large language models show limitations in handling EHR data. While these models can handle some basic algorithms, they currently do not provide sufficient support for the complex predictive analysis required for EHR data.

To address the challenges of medical data analysis, we have adopted a functional decoupling strategy, separating the key functions of EHR data analysis from the system and designing the API standard within the tool bank specifically for the core steps and requirements of EHR data analysis with built-in basic data analysis knowledge and functions. On this basis, we build the specialized agent ToolAgent to understand which tools in the tool bank are helpful for targeted solutions. CodeAgent calls these functions with glue code to smoothly execute them.

In addition, to meet professional physicians' customized advanced EHR data analysis requirements, besides various built-in medical data analysis tools and APIs in the tool bank, it is also easy to register customized use cases into the tool bank by providing related code and instructions for their usage.

### 4 CASE STUDIES

We provide two case studies: Figure 3 showcases how EHRFlow successfully calculates the male and female ratio in a specific dataset. Figure 4 illustrates a more advanced scenario that plots the histogram of patient feature importance, which includes the training of a complex deep learning model. These two cases demonstrate EHRFlow's superior task-solving ability in complicated tasks, while only natural language is input to EHRFlow. The detailed version of the case studies is provided in the Appendix.

### 5 FUTURE WORK

This study provides a preliminary exploration of the application of large language models in the analysis of structured medical data, focusing on EHR data and prognostic prediction tasks of interest to doctors, and systematically designs and implements relevant technologies. The research not only provides a new perspective for the field of medical data analysis but also demonstrates the potential of large language models in processing medical data. However, given the complexity of the field and the continuous advancement of technology, future research needs to delve deeper into and refine aspects such as application-oriented process decomposition, user interaction experience optimization, security and privacy protection, and multimodal data processing. This includes clarifying the boundaries of technical capabilities, expanding system functions, optimizing user engagement processes, innovating user interface design, strengthening data security and privacy protection measures, and developing algorithms and models that integrate multimodal data to enhance the overall performance and clinical application value of the system.

### 6 CONCLUSIONS

This study develops a user-friendly electronic health record (EHR) data analysis platform powered by a large language model. With only natural language interaction, physicians could conduct complex data analysis tasks in code-free manner. EHRFlow applies a multi-agent collaborative mechanism, achieving fine-grained task decomposition and dynamic execution based on user needs, as well as deep customization and optimization of EHR data. The system design is user-oriented, allowing users to express personalized needs through natural language interaction and provide users with an intuitive and flexible data analysis experience. EHRFlow showcases the application potential and practical value of large language models in the field of medical data analysis.

### 7 DISCUSSION ON ALIGNING EHRFLOW WITH AIDSH WORKSHOP SCOPE

The "Artificial Intelligence and Data Science for Healthcare (AIDSH)" workshop is at the forefront of exploring the integration of AI and DS within the healthcare sector, with a particular focus on bridging the gap between data-centric AI and people-centric healthcare (PCHC). The workshop is dedicated to fostering innovative applications of AI and DS that not only advance healthcare practices but also actively engage individuals in managing their health data.

Our paper, EHRFlow, aligns with the AIDSH workshop's research scope in several ways:

- **Enhancing Clinical Decision-Making**: EHRFlow is designed to empower physicians with an intuitive platform that enables them to harness the power of AI through natural language interactions. By facilitating the analysis of EHR data, EHRFlow supports personalized treatments, disease prediction, and clinical research, which are in line with the workshop's goals of driving healthcare innovation with AI and data science.
- **Advancing People-Centric Healthcare (PCHC)**: EHRFlow embodies the PCHC philosophy by integrating digital agents into clinical workflows through interactive conversations. It addresses the skill gap between physicians and data scientists by

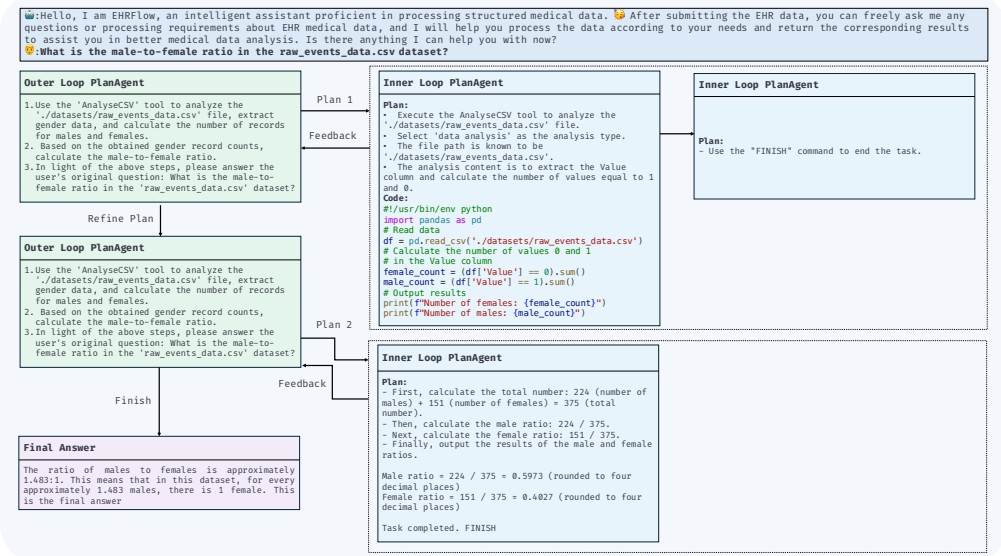

Figure 3: **EHRFlow** successfully calculates the male and female ratio in a specific dataset.

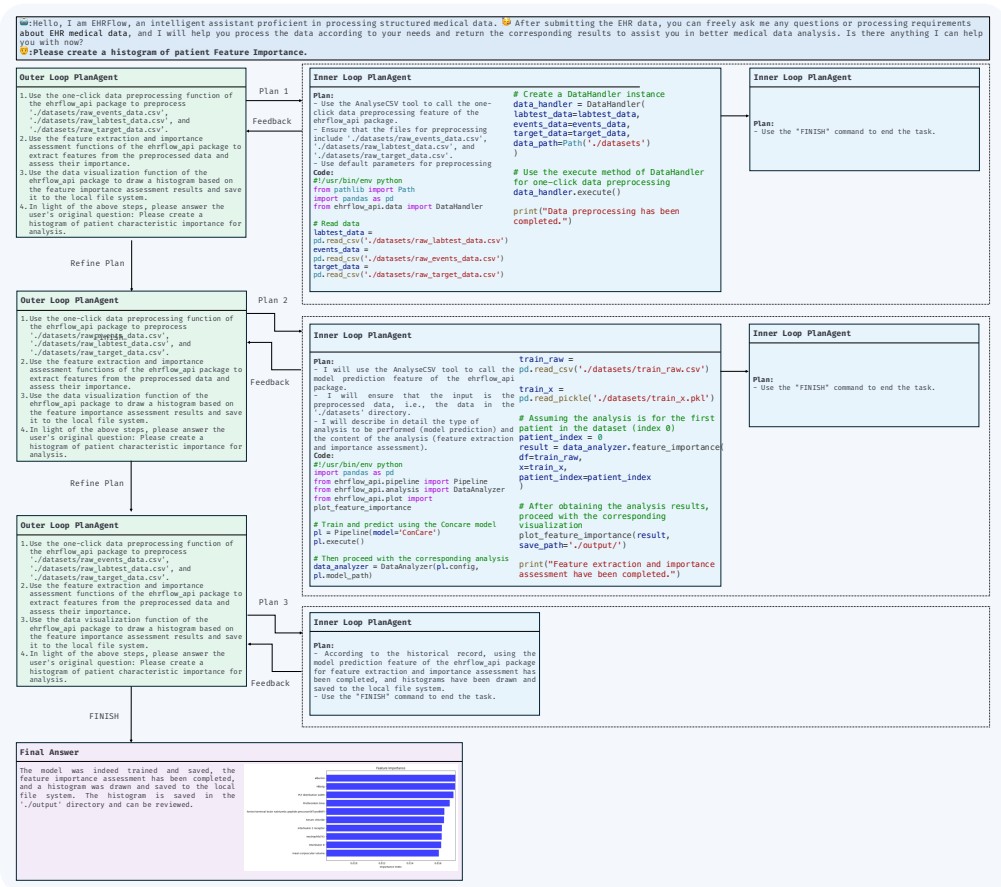

Figure 4: **EHRFlow** successfully plots the histogram of patient feature importance.

providing a platform that requires no coding knowledge. This user-friendly interface encourages collaboration by enabling physicians to specify their data analysis needs directly, thereby promoting active participation in the clinical process.

- **Education Support**: By making the code for the tool banks and `EHRFlow` publicly available, we provide transparent and reliable resources used for educational purposes. This aligns with the workshop's emphasis on data education and support, empowering healthcare professionals to leverage AI and data science in practices.

- **Privacy Protection**: `EHRFlow`'s approach to privacy protection, where only data schema are provided and all sensitive operations are executed locally, aligns with the growing concern for data security in healthcare. This commitment to privacy is a critical aspect of the workshop's focus on responsible AI applications.

In summary, `EHRFlow` not only contributes to the advancement of EHR data analysis but also fits seamlessly within the research scope of the AIDSH workshop. It represents a significant step towards realizing the vision of data-centric AI that is also people-centric, providing a powerful, accessible, and privacy-conscious tool to enhance clinical workflows and decision-making.

## ACKNOWLEDGMENTS

This work was supported by the National Natural Science Foundation of China (U23A20468), and Xuzhou Scientific Technological Projects (KC23143). Junyi Gao acknowledges the receipt of studentship awards from the Health Data Research UK-The Alan Turing Institute Wellcome PhD Programme in Health Data Science (grant 218529/Z/19/Z).

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

## A    WEB INTERFACE OF EHRFLOW

The web interface of EHRFlow is designed to be intuitive and user-friendly, allowing users to seamlessly interact with the model. It features a straightforward data upload mechanism and an input box where users can specify their requirements, offering a user experience akin to GPT-4 (see Figure 5).

## B    DETAILED VERSION OF CASE STUDIES

We provide a comprehensive version of case studies using EHRFlow. These detailed and complete versions delve deeper into the EHRFlow's inner loop and outer loop process, offering a richer understanding of EHRFlow (see Figures 6 and 7).

## C    DETAILS OF THE TOOL BANK

### C.1    Development of EHRFlowAPI

We first designed the EHRFlowAPI, a set of customized interfaces tailored to the specific needs and processes of medical data analysis, aimed at providing bespoke solutions for EHR data processing and analysis. The modular design of EHRFlowAPI allows each module to focus on specific tasks within the data processing and analysis workflow, covering the entire process from data preprocessing, feature extraction, model training and prediction, to data analysis interpretation and result visualization [8, 22]. The development of EHRFlowAPI emphasizes simplifying the process of invoking complex models, particularly to meet the needs of core user groups such as frontline doctors who typically lack professional data analysis and programming knowledge. Therefore, ease of use is a key focus in the API design, with default values set for interface parameters to ensure smooth operation even if users do not provide detailed data analysis parameters. Additionally, the careful design and integration of the interfaces ensure continuity in the data analysis process, allowing users to complete the entire workflow smoothly even with incomplete parameters.

To further enhance operational convenience, we have developed a one-click execution interface that allows users to automatically complete a series of fundamental steps, including data preprocessing and analysis, with a single operation. This greatly lowers the technical barrier and improves the efficiency of data analysis. These design philosophies collectively ensure that users without a technical background can easily use the system for data analysis, while reducing the workload on the model during interface calls, significantly enhancing the user experience.

Through EHRFlowAPI, the role of ToolAgent shifts from generating complete analysis and prediction code to invoking these interfaces. This transition not only reduces the complexity of tasks handled by the large model but also improves the efficiency and executability of code generation, thereby optimizing the overall data analysis process.

### C.2    Construction of the ToolAgent

After developing APIs that cater to the needs of medical data analysis and the characteristics of EHR data, we faced the challenge of enabling large language models to effectively understand and utilize these APIs. Directly inputting code into the model not only occupies a substantial amount of token space but also risks introducing unnecessary redundant information, hindering effective API invocation. To address this issue, we adopted an innovative approach: using natural language to describe the functionality, parameter requirements, and expected return results of the API in detail, instead of directly providing code. Utilizing the GPT-4 model, we generated natural language descriptions of the APIs, emphasizing clearly defining the role and parameter options of each method to ensure that the large language model accurately understands the purpose and operation of the code. Furthermore, we introduced a manual review process to verify the accuracy of the natural language descriptions, eliminating ambiguities and ensuring the model can precisely invoke the API interfaces, thereby enhancing the system's accuracy, reliability, applicability, and safety.

The EHRFlowAPI is designed to meet the core needs of EHR analysis and offers a rich set of interfaces. Despite the exhaustive natural language descriptions of these interfaces, the volume of textual information remains substantial. To manage this, after developing the EHRFlowAPI functionality prompt library based on GPT-4 generation and manual review, we implemented a filtering strategy, providing the model only with descriptions directly relevant to the current task. This ensures that the model efficiently and accurately processes the necessary information. In constructing the EHRFlow system, we established a standardized data analysis workflow: "data preprocessing—model prediction—data analysis—result visualization" to ensure effective API invocation and data processing accuracy. We also built a case library of medical data analysis, containing validated typical cases that cover various aspects of medical data analysis, ensuring the standardization and completeness of the code. ToolAgent can retrieve and integrate professional medical information based on user requirements, execute case and API information, and generate corresponding code solutions, enhancing the coherence of the analysis and execution process and the accuracy of code generation.

Addressing the limitations of large language models in handling long texts and maintaining contextual coherence, we developed a Python code interpreter tool integrated with Jupyter Notebook's interactive design. This tool supports seamless linkage and effective reuse of data and intermediate variables across steps while maintaining context history, significantly improving the precision and efficiency of analysis results. Through these comprehensive measures, we leverage the advantages of large language models in natural language processing while ensuring their efficiency and robustness in practical applications.

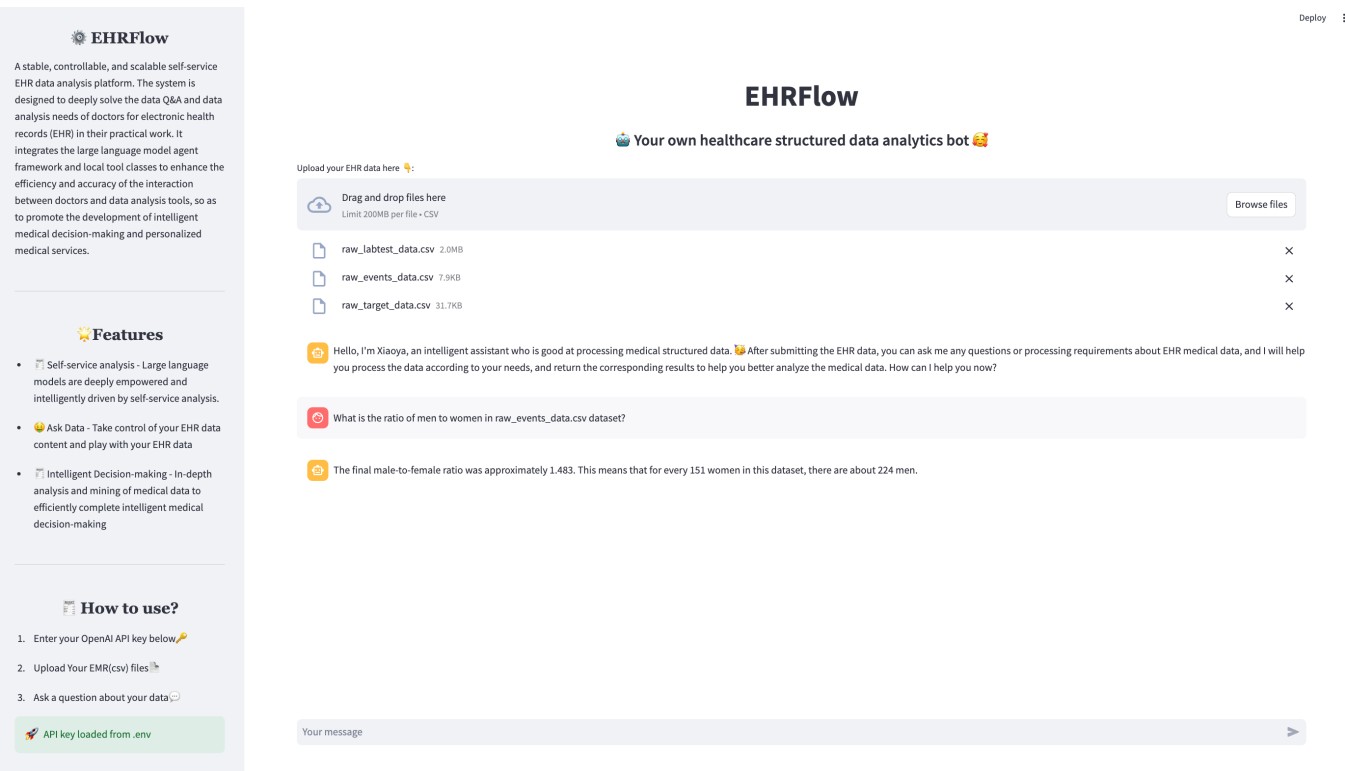

**Figure 5: Web interface of `EHRFlow`. Users first upload the data to be analyzed, then can type their requirements in the input box with a user experience similar to GPT-4.**

EHRFlow: A Large Language Model-Driven Iterative Multi-Agent Electronic Health Record Data Analysis Workflow

👩:Hello, I am EHRFlow, an intelligent assistant proficient in processing structured medical data. 🙂 After submitting the EHR data, you can freely ask me any questions or processing requirements about EHR medical data, and I will help you process the data according to your needs and return the corresponding results to assist you in better medical data analysis. Is there anything I can help you with now?
🧑:What is the male-to-female ratio in the raw_events_data.csv dataset?

**Outer Loop: Planner**

> Entering new EHRFlow chain...
Problem restatement: The user needs to know the male-to-female ratio in the 'raw_events_data.csv' dataset. This means we need to calculate the number of records with male and female genders in the dataset and calculate their ratio.

Key concepts:
- Male-to-female ratio: The ratio of the number of records with male gender to the number of records with female gender.

Concept breakdown:
- Male-to-female ratio
  - Number of records with male gender
  - Number of records with female gender

Reflection:
- We already know that we need to find gender-related records from the 'raw_events_data.csv' file.
- We have not yet obtained the number of records with male and female genders.
- From the current information, we cannot get the specific male-to-female ratio.

Thinking:
- First, we need to extract gender data from the 'raw_events_data.csv' file.
- Then, we need to calculate the number of records with male and female genders.
- Finally, we will divide these two numbers to get the male-to-female ratio.

Reasoning:
- We can use the 'AnalyseCSV' tool to analyze the 'raw_events_data.csv' file, extract gender data, and calculate the number of male and female records.
- Based on these numbers, we can calculate the male-to-female ratio.

&Plan&
1. Use the 'AnalyseCSV' tool to analyze the './datasets/raw_events_data.csv' file, extract gender data, and calculate the number of records with male and female genders.
2. Based on the obtained gender record numbers, calculate the male-to-female ratio.
3. In view of the above steps, please answer the user's original question: What is the male-to-female ratio in the 'raw_events_data.csv' dataset?

**The current plan is set as follows:**
1. Use the 'AnalyseCSV' tool to analyze the './datasets/raw_events_data.csv' file, extract gender data, and calculate the number of records for males and females.
2. Based on the obtained gender record counts, calculate the male-to-female ratio.
3. In light of the above steps, please answer the user's original question: What is the male-to-female ratio in the 'raw_events_data.csv' dataset?

**Inner Loop: Planner**

Now I will proceed to solve this step: Use the 'AnalyseCSV' tool to analyze the './datasets/raw_events_data.csv' file, extract gender data, and calculate the number of records with male and female genders.
>>>>Round: 0<<<<

### Thinking process
Current task: Use the 'AnalyseCSV' tool to analyze the './datasets/raw_events_data.csv' file, extract gender data, and calculate the number of records with male and female genders.

Key concepts:
- Gender data: In the './datasets/raw_events_data.csv' file, gender data is represented by the Value column, where male is 1 and female is 0.

Concept breakdown:
- Gender data
  - Extract data from the Value column
  - Calculate the number of values equal to 1 (number of males)
  - Calculate the number of values equal to 0 (number of females)

Reflection:
- From the historical record information, it is already known that the './datasets/raw_events_data.csv' file contains gender data, and gender data is stored in the Value column.
- The task requires calculating the number of records with male and female genders, which means counting the data in the Value column.
- The specific number of gender data has not yet been obtained.
- The AnalyseCSV tool needs to be used to analyze and calculate the gender data.

Thinking:
- Need to first extract the data from the Value column, then calculate the number of males and females based on the values.
- The extraction and calculation tasks can be completed in one step, without the need to separate them.
- The AnalyseCSV tool can accomplish this task, specifying the analysis type as 'data analysis'.
- There is no need to assume the distribution of gender data, just get the actual values directly from the data file.

Reasoning:
- Use the AnalyseCSV tool, specify the analysis type as 'data analysis', the file path as './datasets/raw_events_data.csv', and the analysis content as extracting the Value column and calculating the number of males and females.

Plan:
- Execute the AnalyseCSV tool to analyze the './datasets/raw_events_data.csv' file.
- Select 'data analysis' as the analysis type.
- The file path is known to be './datasets/raw_events_data.csv'.
- The analysis content is to extract the Value column and calculate the number of values equal to 1 and 0.

**Execute action**

```json
{
"name": "AnalyseCSV",
"args": {
"inputs": {
"type": "data analysis",
"files": ["./datasets/raw_events_data.csv"],
"content": "Extract data from the Value column, calculate the number of values equal to 1 (number of males) and the number of values equal to 0 (number of females)."
}
}
}
```

```AnalyseCSV
#!/usr/bin/env python
import pandas as pd
# Read data
df = pd.read_csv('./datasets/raw_events_data.csv')
# Calculate the number of values 0 and 1
# in the Value column
female_count = (df['Value'] == 0).sum()
male_count = (df['Value'] == 1).sum()
# Output results
print(f"Number of females: {female_count}")
print(f"Number of males: {male_count}")
```

**Code Reviewer**

type=True content='Code review passed. The code structure is clear and meets the specification requirements. Variable naming is standardized and comments are sufficient. The code correctly implements the user's requirements and can calculate and output the number of males and females. Exception and error handling are not reflected in the code snippet, but according to the provided code, there is no obvious need for exception handling. The code can be run and no syntax errors or dependency issues have been found.'

**Execute Result**

Number of males: 224
Number of females: 151

**Inner Loop: Planner**

> > > > Round 1 < < < <
The thought process

Current task: Use the 'AnalyseCSV' tool to analyze the './datasets/raw_events_data' file, extract gender data, and calculate the number of records with gender male and gender female.

Key concepts:
- Gender data: In the './datasets/raw_events_data' file, gender data is represented by the Value table, where males are 1 and females are 0.

Concept teardown:
- Gender data
  - Extract data from the Value column
  - Calculate the number of values of 1 in the Value column (number of males)
  - Calculate the number of values of 0 in the Value column (number of women)

Reflection:
- From the history information, it is known that the './datasets/raw_events_data' file contains gender data and that gender data is stored in the Value column.
- The task needs to count the number of records whose gender is male and female, which means that the data in the Value column needs to be counted.
- According to the previous execution results, the number of records with male gender and female gender has been 224, and with female gender is 151.
- The current task has been completed and should be completed using the "FINISH" command.

Think:
No further thought is required because the task has been completed.

Reasoning:
No further reasoning is required because the task has been completed.

Plan:
- Use the "FINISH" command to end the task.

**Execute action**

```json
{
"Name": "FINISH"
}
"FINISH
```

**Execute Result**

---
FINISH

**Outer Loop: Planner**

**Reflection**

Step: value = "Use the'AnalyseCSV 'tool to analyze the './datasets/raw_events_data 'file to extract gender data and calculate the number of records with male and female gender."

Response: Based on the information provided above and the code executed, the final answer is:

Number of men: 224
Number of women: 151

This means that in the './datasets/raw_events_data .csv' file, there are 224 records of gender as male and 151 records of gender as female. Based on the information provided above and the code executed, the final answer is:

Number of men: 224
Number of women: 151

This means that in the './datasets/raw_events_data .csv' file, there are 224 records of gender as male and 151 records of gender as female.

**Refine Plan**

The current plan is set to:
1. Use the'AnalyseCSV 'tool to analyze'./datasets/raw_events_data 'file, extract gender data and calculate the number of records with gender as male and gender as female. -> 2. Calculate the ratio of the number of males to the number of females. -> 3. Given the above steps, please answer the user's initial question: What is the ratio of males to females in the 'raw_events_data .csv' dataset?

**Inner Loop: Planner**

>>>>Round: 0<<<<

### Thinking Process

#### Current Task:
The task that needs to be addressed currently is to calculate the ratio of the number of males to the number of females.

#### Key Concepts:
- Number of males: In the previous task, it was determined that there are 224 males.
- Number of females: In the previous task, it was determined that there are 151 females.

#### Concept Breakdown:
- Ratio Calculation:
  - Male ratio = Number of males / Total number
  - Female ratio = Number of females / Total number
  - Total number = Number of males + Number of females

#### Reflection:
- The values for the number of males and females have been obtained from historical records and do not need to be recalculated.
- The total number can be obtained through simple addition.
- Ratio calculation is a straightforward mathematical problem that can be directly calculated.

#### Consideration:
- The total number must be calculated first before the ratios of males and females can be determined.
- Since the number of males and females is known, direct calculation can be performed without querying the data source.
- Calculating the ratio is a simple mathematical operation that does not require the use of complex tools or commands.

#### Reasoning:
- Based on the known number of males and females, the total number can be directly calculated.
- With the total number, the ratios for males and females can be determined.

#### Plan:
- First, calculate the total number: 224 (number of males) + 151 (number of females) = 375 (total number).
- Then, calculate the male ratio: 224 / 375.
- Next, calculate the female ratio: 151 / 375.
- Finally, output the results of the male and female ratios.

**Execute action**

```json
{
"name": "FINISH",
"args": {}
}
```

Since the current task is to calculate the ratio, which is a simple mathematical calculation, no tools or commands are needed. Therefore, I will directly provide the calculation results and end the task using the "FINISH" command.

Male ratio = 224 / 375 = 0.5973 (rounded to four decimal places)
Female ratio = 151 / 375 = 0.4027 (rounded to four decimal places)

Task completed. FINISH

**Execute Result**

---
FINISH
*****

**Outer Loop: Planner**

**Reflection**

Step: value='Calculate the ratio of the number of males to the number of females.'

Response: Based on the information provided in the previous task, the number of males is 224 and the number of females is 151. To calculate the ratio of the number of males to the number of females, we can divide the number of males by the number of females.

Ratio of males to females = Number of males / Number of females
Ratio = 224 / 151

Upon calculating this ratio, we get:

Ratio = 1.483

So, the ratio of males to females is approximately 1.483:1. This means that in this dataset, for every approximately 1.483 males, there is 1 female. Based on the information provided in the previous task, the number of males is 224 and the number of females is 151. To calculate the ratio of the number of males to the number of females, we can divide the number of males by the number of females.

Ratio of males to females = Number of males / Number of females
Ratio = 224 / 151

Upon calculating this ratio, we get:

Ratio = 1.483

So, the ratio of males to females is approximately 1.483:1. This means that in this dataset, for every approximately 1.483 males, there is 1 female. This is the final answer.

**Final Answer**

👩:{'input': 'What is the male-to-female ratio in the raw_events_data.csv dataset?', 'output': 'Based on the information provided in the previous task, the number of males is 224 and the number of females is 151. To calculate the ratio of the number of males to the number of females, we can divide the number of males by the number of females.

Ratio of males to females = Number of males / Number of females
Ratio = 224 / 151

Upon calculating this ratio, we get:

Ratio = 1.483

So, the ratio of males to females is approximately 1.483:1. This means that in this dataset, for every approximately 1.483 males, there is 1 female. This is the final answer.'}

**Figure 6: EHRFlow successfully calculates the male and female ratio in a specific dataset. (Complete version)**

**Figure 7: `EHRFlow` successfully plots the histogram of patient feature importance. (Complete version)**