# OpenReview forum: "EHRFlow: A Large Language Model-Driven Iterative Multi-Agent Electronic Health Record Data Analysis Workflow"
_KDD.org/2024/Workshop/AIDSH — KDD-AIDSH 2024 Oral_

### Official Review · Reviewer_ZRCC · 2024-06-17
**EHRFlow, a user-friendly electronic health record (EHR) data analysis platform powered by a large language model**

**Rating:** 9
**Confidence:** 4

**Review:**

This study developed a user-friendly electronic health record (EHR) data analysis platform,  EHRFlow, powered by a large language model. EHRFlow showcases the application potential and practical value of large language models in the field of medical data analysis. This work has excellent quality, clarity, originality and significance and not only contributes to the advancement of EHR data analysis but also fits seamlessly within the research scope of the AIDSH workshop. However, there are two comments and suggestions for revisions regarding this research.
1. The citation numbers of the references and image numbers need to be reordered in the order they appear in the main text.
2. The URL, https://anonymous.4open.science/r/EHRFlow, can not open.

---

### Official Review · Reviewer_o3qb · 2024-06-17
**Evaluation of "EHRFlow: A Large Language Model-Driven Iterative Multi-Agent Electronic Health Record Data Analysis Workflow"**

**Rating:** 7
**Confidence:** 4

**Review:**

This paper presents a large language model-driven platform that simplifies electronic health record (EHR) data analysis for physicians through natural language interactions. The methodology is detailed and the integration of various agents to manage different tasks shows a robust and innovative approach. My major comments are:
1.	 This paper may be improved using a more detailed evaluation of platform's performance. Can you provide a method to validate your results?
2.	Can you also provide a more detailed comparison with existing healthcare data analytic tools?
Overall, the work has the potential to greatly enhance efficiency and effectiveness of clinical workflows.

---

### Decision · Program_Chairs · 2024-06-28

Accept (Oral)